# Latex Allergens in the Ear Straps of Face Coverings and Masks in the UK: Initial Findings

**Howard Mason *** [ID] **and Iwona Rosa**

The Science and Research Centre, Health & Safety Executive, Buxton SK17 9JN, UK; iwona.rosa@hse.gov.uk
* Correspondence: howard.mason@hse.gov.uk; Tel.: +44-7771743441

**Abstract:** The SARS-CoV-2 pandemic led to an unprecedented demand for PPE and generated a small-scale industry making personalised face coverings. Concerns had been raised about the use of natural rubber latex (NRL) as an elastomeric material, with its health risks. We have investigated the levels of four NRL allergens and total protein in elastomeric ear-straps in manufactured face coverings and the material sold for their production, and a number of imported N95/KN95 PPE masks. None of the samples identified whether NRL was involved or not. NRL allergens levels in manufactured masks were low or not detectable; 3/10 of the N95/KN95 masks showed levels above the limit of detection, probably reflecting low-level cross-contamination during manufacture. Three batches of material sold for "the manufacture of ear straps for face coverings" had significant but variable levels of allergen (250–2526 ng/g of material). Historically, extractable protein measurements have been used as an indicator of possible NRL proteins. This study showed significant levels of apparent protein in sample extracts without measurable NRL allergens or confirmation by electrophoresis. Therefore, the immunochemical measurement of NRL allergens remains key to rule out elastomeric material with the potential to cause latex-related health problems.

**Keywords:** PPE; face coverings; natural rubber latex; allergens

## 1. Introduction

The SARS-CoV-2 (COVID pandemic is estimated to have led to 22.6 million cases and caused over 180,000 deaths in the UK. The early realisation of the key role of airborne transmission of the virus led to a sustained, very high demand for Personal Protective Equipment (PPE), especially face masks. While this led to innovative approaches to both the manufacture and incorporation of novel materials for enhanced protection in PPE [1,2], such innovations need regulatory approval before entering the mainstream PPE market. The immediate international demand for PPE caused significant problems to global supply chains [3], with much of the PPE being manufactured in China. The UK government was committed to not only procuring PPE from such existing suppliers, but also aimed to encouraging new, UK-based entrants into PPE manufacture and ordered some 32 billion items of PPE from February to July 2020 against the background of intense international competition [4,5].

Moreover, as in many countries, the UK general population was extolled or mandated to wear "face coverings" in certain normal life activities. This further increased demand for masks, together with an ad hoc production of face coverings. The latter did not purport to meet any certified standards, but often with a desire for their individualisation in terms of material and decorative patterns and produced by "cottage industries" or larger scale manufacture.

Masks or face coverings are often secured by elastomeric straps using the ears to ensure secure anchorage and a good fit around the nose and mouth. While a range of elastomeric materials is available, the properties and availability of natural rubber latex (NRL) from

*Hevea brasiliensis* means that it may be employed in this context without the necessary consideration of its well-known allergenic potential during manufacture or end-use.

Several potential health effects are identified with NRL products. Individuals sensitised to latex can have an IgE-mediated response to NRL allergens, causing skin and asthma-like respiratory symptoms; symptoms can be severe and systemic. The average prevalence of latex allergy in the general population and healthcare workers has been reported as 1–4% and 10%, respectively [6]. There is also some cross-reactivity between allergens in NRL and those found in some fruit species [7]. NRL products can also cause type IV cell-mediated contact allergic dermatitis, with symptoms usually limited to the site of skin contact and taking up to 48 h to develop, and non-allergic contact dermatitis. Both forms of contact dermatitis are usually caused by chemicals used in the manufacture and processing of NRL products, and through extended skin contact. There are concerns that shortening or hurrying the production of NRL products can lead to increases in residual allergens and chemicals in the product.

Allergic reactions to NRL have been well documented [8], very largely in the context of medical/surgical gloves [9,10]. The HIV/AIDS epidemic in the late 1980s is considered to have led to a significant increase in health workers becoming sensitised to NRL allergens from increased use of surgical gloves, and possibly increased residual allergens and chemicals due to the excessive supply pressures on manufacturers [11]. Subsequently, there was a move to non-latex or non-powdered latex gloves with low or non-detectable allergen content. However, workplace respiratory and dermal health issues from manufacturing processes involving NRL in the textile industry have also been reported [12–14].

A number of NRL allergens have been identified, and standard methods for determining the NRL allergen content of latex products have been produced [15,16]. Four of the major NRL allergens, Hev b 1, Hev b 3, Hev b 5 and Hev b 6.02 have commercially available immunoassays.

This work was initiated due to concerns brought to my attention about possible NRL allergen involvement in two health workers who reported dermal facial lesions and used both hospital-supplied PPE and personalised face coverings. There appeared to have been no published evidence on whether the unprecedented demand for PPE and face coverings during the COVID pandemic had led NRL being used as a component of facemasks for professional use or face coverings for the public. The main purpose of this work was to identify whether NRL allergens could be identified in elastomeric ear straps taken from a small range of face covering and masks produced during the COVID pandemic. A secondary purpose, given the cost of the NRL allergen immunoassays, was to investigate whether a simple, readily available protein assay could be used as an initial screening approach for possible NRL involvement.

## 2. Materials and Methods

Materials being sold for use as ear straps in the production of face coverings and the ear straps from manufactured masks/face-coverings imported into the UK were obtained from a variety of sources. Samples 1–5, 8–10, 12–14 were available from online resources geared towards smaller-scale production of face coverings during the pandemic. Samples 13 and 14 were ordinary stationers' elastic bands, which early in the pandemic was suggested on the internet as useful for ears straps, although previously recognised as a potential source of NRL-related problems [17]. Samples 6, 7 and 11 were manufactured masks/face coverings that were offered for sale either online or widely available for sale to the public. Table 1 identifies the characteristics of samples 1–14.

**Table 1.** Results for the 15 samples sold for use as ear straps in the production of face coverings and the ear straps from manufactured face-coverings for sale to the general public.

| | Description | Hev b 1 ng/g | Hev b 3 ng/g | Hev b 5 ng/g | Hev b 6.02 ng/g | ∑ 4 Hev ng/g | Protein µg/g |
|---|---|---|---|---|---|---|---|
| 1 | Black flat elastic, sampled from 2 m length sold for attachment on cloth face coverings # | 466 | 177 | 40 | 43 | 726 | 997 |
| 2 | White round elastomeric cord, sampled from 5 m length sold for attachment to face coverings # | ND | ND | ND | ND | ND | 4486 |
| 3 | Pre-cut flat soft elastic blue cord, with adjustable toggles for attachment to face coverings (toggles removed) | ND | ND | ND | ND | ND | ND |
| 4 | Pre-cut flat soft elastic red cord with adjustable beads for attachment to face coverings (beads removed) | ND | 13 | 27 | ND | 40 | 301 |
| 5 | 3 mm round elastometric cord, sampled from a 10 m length. Sold for attachment for cloth face coverings # | ND | ND | ND | ND | ND | 2355 |
| 6 | Ear straps sampled from 2 of a pack of 10 disposable, non-medical masks, origin China # | ND | ND | ND | ND | ND | 1356 |
| 7 | Ear straps sampled from 2 of a pack of 3 disposable masks marked EN14683, origin China | ND | ND | 19 | ND | 19 | ND |
| 8 | Flat knitted elastic, sampled from 5 m length. Sold for attaching as ear straps to cloth face coverings | 219 | 13 | 18 | ND | 250 | ND |
| 9 | Pack of pre-cut white flat elastic with toggles for attachment to face coverings (toggles removed) | ND | ND | ND | ND | ND | ND |
| 10 | Round elastomeric material sold for making face coverings, origin China. Sampled from a 10 m length # | ND | ND | ND | ND | ND | 3463 |
| 11 | Commercial washable cloth black face covering, sold singly | ND | ND | ND | ND | ND | 43 |
| 12 | Grey elastic flat tape, sampled from 5 m length. Sold for ear attachments on cloth face coverings | 1240 | 377 | 98 | 811 | 2526 | 460 |
| 13 | Pack of new elastic bands. Early in the pandemic online sources suggested could be used for ear straps | ND | 280 | ND | 9 | 289 | ND |
| 14 | Second pack of new elastic bands. As above, but from a different supplier | ND | 258 | ND | ND | 257 | 4 |

ND—not detected or below the limit of detection. Hevs b 1 b 2, b 3, b 5 and 6.02 quantified by ELISA; protein quantified by BCA protein assay. # indicates samples that underwent gel electrophoresis.

A total of 10 of the samples (identified as samples 15–24, Table 2) were from masks imported into the UK under government contract for professional use and certified as meeting a number of standards, specifically N95/KN95 standards; 1 of these 10 masks (sample 19, Table 1) was described as a medical protective mask meeting standard GB19083-2010.

**Table 2.** Ten samples of ear straps from masks imported into the UK under government contract for professional use and certified as meeting N95/KN95 standards.

| | Description | Hev b 1 ng/g | Hev b 3 ng/g | Hev b 5 ng/g | Hev b 6.02 ng/g | ∑ 4 Hev ng/g | Protein µg/g |
|---|---|---|---|---|---|---|---|
| 15 | Professional FFP2 mask, origin China. Certified GB2626-2006/EN-149-2001+A1-2009/KN95/CE. | ND | ND | ND | 17 | 17 | 750 |
| 16 | Professional FFP2 mask, origin Switzerland. Certified EN-149-2001+A1-2009/CE 2834 | ND | ND | ND | ND | ND | ND |
| 17 | Professional FFP3 mask, non-medical, origin China. Certified GB2626-20006/EN-149-2001+A1-2009/CE # | ND | ND | ND | ND | ND | 1850 |
| 18 | Professional mask, origin China. Certified KN95/GB2626-2006/CE | ND | 11 | 16 | ND | 27 | ND |
| 19 | Professional mask, origin China. Certified GB19083-2010 medical protective masks standards | 31 | 11 | 16 | ND | 58 | 708 |
| 20 | Professional mask, origin China. Certified EN-149 2001+A1-2009/CE | ND | ND | ND | ND | ND | ND |
| 21 | Professional mask, origin China. Certified GB2626-2006/KN95 | ND | ND | ND | ND | ND | 1753 |
| 22 | Professional mask, origin China. Certified KN95/EN-149-2001+A1-2009/CE | ND | ND | ND | ND | ND | 2922 |
| 23 | Professional mask, origin China. Certified KN95/GB2626-2006/CE | ND | ND | ND | ND | ND | 2965 |
| 24 | Professional FFP2 mask, origin China. Certified KN95/EN-149-2001+A1-2009 | ND | ND | ND | ND | ND | 1903 |

ND—not detected or below the limit of detection. Hevs b 1 b 2, b 3, b 5 and 6.02 quantified by ELISA; protein quantified by BCA protein assay. # indicates samples that underwent gel electrophoresis.

A minimum of 1 g was sampled, comprising material from multiple items, e.g., both ear straps from two masks; multiple cuts of elastic from a single length. These were extracted for analysis of four latex allergens and total protein. Extraction of a known weight of elastic material was carried out at 10% $w/v$ using phosphate-buffered saline containing 0.1% Tween 20 (Merck, Gillingham, UK). Extractions involved mixing over-end overnight at room temperature. Subsequently the extracts were centrifuged at $3000\times g$ for 20 min at 8 °C, the supernatants were then passed through 0.45 micron filters and subsequently stored at −20 °C until analysis.

Hev b 1, Hev b 3, Hev b 5 and Hev b 6.02 were measured by commercial, non-competitive sandwich immunoassay (FITkit Icosagen, Tartu, Estonia) on an automated ELISA platform (Triturus, Grifols, Cambridge, UK). Total protein was measured using the bicinchoninic acid (BCA) protein assay (Thermo Scientific, Warrington, UK). All measurements were performed in duplicate.

Results below the limit of detection (LOD) for the immunoassays and protein measurements were defined as non-detected (ND). LODs were estimated using the ProQuant software (QIVX Inc, Fort Collins, US), as the concentration corresponding to the response at zero dose plus 3 times the pooled standard deviation.

In order to identify the nature of any protein identified in the total protein assay, five extracts showing protein concentrations greater than 0.2 mg/mL underwent SDS gel electrophoresis in duplicate on a 12% BisTris gel and MES buffer system (Biorad, Watford,

UK). Loading amounts were 14–43 micrograms per lane and an irrelevant 18 kDa rabbit protein was loaded at 18 micrograms for comparative purposes. The gel was stained with Gel Code Blue Coomassie stain (Thermo Scientific, Warrington, UK). After de-staining, visible protein bands from the extracts were excised for protein identification (University of York Proteomics laboratory, York, UK) by LC-MS/MS after trypsin digestion, and peptides subsequently matched using the Mascot search engine against a number of databases. These included Uniprot, as well as the Common Repository of Adventitious Proteins, (cRAP) which list proteins commonly found in proteomics experiments that are present either by accident or through unavoidable contamination of protein samples.

Duplicate lengths (0.3 m) of two samples (1 and 12) that showed high levels of the summed NRL allergens were included with 5 kgs of clothing to a standard 40 °C washing machine program. Levels of allergens were measured in identical lengths of samples 1 and 12 without washing and after washing. Extractions and analyses were carried out as previously described.

### 3. Results

The LODs for Hev b 1, Hev b 3, Hev b 5 and Hev b 6.02 were calculated as 2.0, 1.0, 1.5 and 1.5 ng/mL, respectively.

Table 1 shows the nature of the samples, the four individual latex allergens and their summed value ($\sum$ 4 Hev) and the levels of extractable total protein. Results were expressed per g of extracted material. Three of the samples sold as "elastic for making face coverings" (Nos: 1, 8 and 12 identified in Table 1) had significant levels of latex allergens, ranging over a 10-fold difference in the summed four allergens (250–2526 ng.g$^{-1}$), but with Hev b 1 predominating. The three manufactured non-medical masks (Nos: 6, 7, 11) that were widely available for the public to purchase showed either non-detected or low levels of latex allergen.

The commercial N95/KN95 masks (samples 15–24) for professional use also had generally low or non-detected levels of the latex allergens, although 3/10 showed detectable latex allergen levels above the LOD (Table 2).

There was no significant correlation between protein and summed allergen levels for all samples (Tables 1 and 2) by rank correlation analysis ($p$ = 0.08).

The electrophoretic gel showed only a very single faint band at around 65–68 kD in all five samples tested in comparison to a similar amount of an irrelevant rabbit protein loaded on the gel (Figure 1). There was no obvious relationship between the amount of protein applied to the gel and the level of staining in these bands. The results from the excision of protein bands and protein proteomic identification for all 5 sample extracts (1, 2, 5, 6, 17) showed human keratin was present in the excised bands for each sample, with K2C1 keratin the predominant form based on the exponentially modified Protein Abundance Index (emPAI) values. K1C9 human keratin was present at lower abundancies in all samples. In the protein band from sample 1 bovine serum albumin (BSA) was identified with a higher abundance from emPAI values than the keratins K2C1 and K1C9.

The mean level of summed NRL allergens in the unwashed lengths of samples 1 and 12 were 795 ng/g and 2730 ng/g, respectively. After the standard wash program, the levels had decreased the summed allergen levels to 108 ng/g and 328 ng/g respectively. Therefore, the total allergen levels in samples 1 and 12 had been decreased by 86–88%.

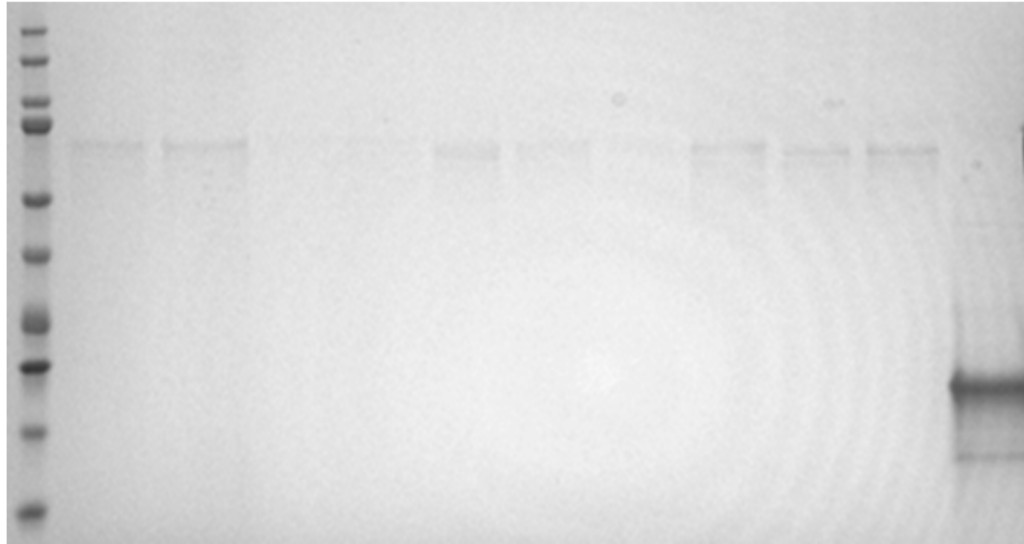

**Figure 1.** SDS reduced gel. Lanes from left to right; molecular weight marker lane (including bands at 10, 15, 20, 25, 37, 50, 75, 100, 150 and 200 kaD); five extracts (samples 1, 2, 5, 6 and 17) in duplicate at loading amounts of 14–43 micrograms per lane and at the extreme right of the gel an 18 kD rabbit protein loaded at 18 micrograms for comparative purposes on the degree of staining.

## 4. Discussion

The summed levels of the four latex allergens were generally found to be low or undetectable in the sampled N95/KN95masks and finished face coverings. The measurable but low levels found in 3/10 of N95/KN95 masks may possibly reflect NRL cross-contamination during the manufacturing process.

None of the products made any statement in English on the packaging as to whether the product did or did not contain NRL. One of the KN95/N95 masks (Sample 24) stated on the box that it was "hypoallergenic" without further clarification. Sample 19, which claimed to meet a medical protective mask standard, stated "do not use if allergic to the materials".

A 10-fold difference in summed allergen content was found in three different batches of elastic being sold for small scale production of face-coverings. UK guidance on the production of face coverings states that allergenic materials, such as latex in elastic, where worn against the skin should be avoided or a warning included on packaging and on the covering itself [18]. This high level of variation in NRL allergens has been reported across similar latex containing products [19]. Small-scale producers of face coverings may have been purchasing significant lengths of NRL elastic without any idea of the levels of allergenic proteins. We have no evidence of airborne levels of allergens during manufacture, subsequent handling, and end-use, posing a respiratory as well as a dermal risk. However, several published papers have identified significant exposures and adverse health effects in the manufacture of elastic ribbon/braiding etc. [12–14].

There is no definitive published data on the relationship between levels of NRL allergens and the risk of poor health outcomes, especially dermal problems. However, in our opinion, it is unlikely that such very low levels of allergens in the three N95/KN95 would cause a health problem, whereas the higher levels found in some components of face coverings may indicate a higher risk of precipitating skin problems.

While there is no evidence that any of the four allergens is more likely to cause health problems, Hev b 3 had been associated with latex sensitisation in spina bifida cases from use of latex catheters, while Hev b 6.02 has tended to be associated with health care workers and their use of latex gloves. The new commercial stationery rubber bands in this study showed only measurable levels of Hev b 3. However, there are reports of contact dermatitis in postal workers from rubber bands [17]. The data in Tables 1 and 2 for samples where NRL was obviously present suggest that the measurement of all four allergens is necessary.

The molecular weights of Hevs b 1, b 3, b 5 and b 6.02 are 14 kDa, 23 kDa, 16 kDa and 4.7 kDa, respectively. There were no bands visible at these molecular weights on the gel. However, given the levels of NRL allergen are measured in ng/mL, and thus picogram amounts would be added to the gel lane, a Commassie-stained gel would not have the necessary sensitivity to detect the presence of NRL allergens. The purpose of the electrophoresis gel was not to detect NRL allergens, but to investigate the nature of the high apparent protein levels found by BCA protein assay in some extracts. The amount of protein staining on the gel did not reflect the apparent protein levels by BCA protein. The proteomic analysis of the only and weakly stained bands at around 65–68 kDa identified keratin contamination in all samples, together with BSA in one of the samples. Low level human keratin from shed skin cells can be a common contaminant, and our laboratory uses significant amounts of BSA. The apparent measurable protein by BCA method in the elastomeric extracts does not actually reflect protein from biological products, such as NRL, but likely some interfering substance(s).

Prior to available immunoassay methods to detect latex proteins and allergens [15,20], a modified Lowry protein method was used as a surrogate of the levels of allergens extractable from latex gloves [21]. The BCA and Lowry methods show some similarities in detecting proteins by a two-stage reaction. First, peptide bonds in proteins reduce $Cu^{2+}$ ions to $Cu^+$ in alkaline solution, then subsequently for the BCA method molecules of bicinchoninic acid chelate with the $Cu^+$ ion, while for the Lowry method the $Cu^+$ reacts with Folin–Ciocalteu reagent. Both forming strongly absorbing chromophores. We conclude that the "apparent protein" from some elastomeric material may suggest that significant reductant is present or unknown extractable substances that react directly with bicinchoninic acid.

The term "elastic" is ill-defined. It may suggest an NRL *Hevea brasiliensis*-based product of variable allergen/protein content, refer to a non-*Hevea brasiliensis* biological material or a chemically synthesised elastomeric material. Although without any guidance as to whether "elasticated" ear straps for producing face coverings may be high in, low in or devoid of NRL allergens, the prudent end-use (or concerned small-scale producer) would have washed the material prior to its use or in producing a face covering. While a single laundry wash at 40 °C reduced any present NRL allergen protein by about eight-fold, there are no definitive "safe" levels, especially for sensitised individuals. Someone who knows or suspects that they are sensitised to latex or to those fruit (e.g., banana, avocado and kiwi) which contain latex cross-reacting proteins [7], needed to take particular care and it would have been preferable for them to wear a face covering with non-elastomeric ties.

Some non-NRL elastomeric material also needs to undergo chemical vulcanisation and other processes in a similar manner to the production of NRL, and therefore have the possibility to cause contact allergic or contact irritant dermatitis. In the UK, reports of adverse reactions to PPE used for medical purposes (i.e., classed as a medical device) are collected by the Medicines and Healthcare products Regulatory Agency and for non-medical PPE by the Health and Safety Executive; both governmental bodies. For face coverings used by the public the route for reporting adverse effects was not so clearly defined. While there have been isolated reports of NRL problems associated with PPE for professional use [22], the limited data presented here suggests that NRL-related problems are more likely to have occurred from the use of face coverings.

## 5. Conclusions

Based on this small study, there was no evidence that the extreme pressure for PPE caused by the COVID pandemic had led to NRL being employed as a component of elastomeric ear straps from masks imported as meeting N95/KN95 standards, or manufactured masks/face coverings for sale to the public. The low levels of NRL allergens found in some of these masks probably indicate low-level cross-contamination during manufacture and are unlikely to pose significant health risks. Several of the elastomeric components advertised for sale or suggested for use as ear straps to produce cloth face coverings for

the public had significantly higher levels of NRL allergens without indication that they contained NRL. Therefore, the risk of NRL-related health issues was more likely to be related to the small-scale production of face coverings.

Immunoassays for four specific NRL allergens are necessary to detect their involvement in elastomeric material.

**Author Contributions:** Conceptualisation, H.M.; Methodology, H.M. and I.R.; Software, H.M.; Validation, H.M. and I.R.; Formal analysis, H.M. and I.R.; Investigation, H.M.; Resources, H.M. and I.R.; Writing—original draft, H.M.; Writing—Review and editing, H.M. and I.R.; Visualisation, I.R.; Supervision, H.M.; Project Administration, H.M. and I.R.; Funding Acquisition, H.M. All authors have read and agreed to the published version of the manuscript.

**Funding:** This publication and the work it describes were funded by the Health and Safety Executive (HSE) under PHF21170, Science Dissemination Program. Both authors are employees of the HSE. Its contents, including any opinions and/or conclusions expressed, are those of the authors alone and do not necessarily reflect HSE policy.

**Institutional Review Board Statement:** Not applicable.

**Informed Consent Statement:** Not applicable.

**Data Availability Statement:** Not applicable.

**Acknowledgments:** The authors wish to thank the Metabolomics & Proteomics laboratory at the Bioscience Technology Facility, University of York, for protein identification.

**Conflicts of Interest:** The authors declare no interests or potential conflict of interest related to this publication. The funders had no role in the design of the study; in the collection, analyses, or interpretation of data; in the writing of the manuscript, or in the decision to publish the results.

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
