# Peer review of "Latex Allergens in the Ear Straps of Face Coverings and Masks in the UK: Initial Findings"

_covid, doi:10.3390/covid2070066_

Round 1

Reviewer 1 Report

The authors have chosen an often overlooked yet important topic to explore.  They have the makings of a nice manuscript that will interest many readers of the journal.

In the manuscript entitled "Latex allergens in the ear straps of face coverings and masks in the UK: initial findings", the authors present their work in regard to detecting natural latex antigens in various face masks/materials.  As the title indicates, their experimentation is limited but appropriate for the stated topic.

Overall, the manuscript is well written, clear, and straight to the point.  Some sections, however, are incomplete or would benefit from additional clarification.  Requested improvements from this reviewer are as follows:

  • What is the purpose of the manuscript?  Initially the focus seemed to be to detect the presence of NRL allergens, yet concluding paragraphs seemed to indicate that the purpose was to determine the best method for detection of NRLs.  Either way, a clear purpose should be chosen/stated.
  • Along with the previous comment, the discussions of the Lowry/BCA method of protein quantification need better integration throughout the text.  (e.g., In the introduction, the Lowry method is not mentioned at all – the only introduction of the method is in the abstract, where it is called the “historical” method.  Please clarify why the Lowry method was used in the context of your work.)  Also, please standardize the name.  In line 101 it is the modified Lowry method, in line 190 it is the BCA method, and every other mention is just the Lowry method.
  • The Materials and Methods section is not complete.
    • Please remove the publisher instructions (lines 64-77).
    • Materials are not consistently described with manufacturer information.
    • On line 96, at what temperature is the centrifugation performed?
    • Line 140 in the Results section mentions a 40C wash procedure that is not described in the Methods section.
  • Question: How relevant are the detectable allergen levels?  Are standard antigenic concentrations known to induce the latex-related health problems?  Is any one allergen more likely to produce health problems than the other 3?  (I did not find the answers with a brief online search, so I cannot be certain that answers are known – but if they are, they should be included in the manuscript!)
  • Be careful to only include observations in the Results.  Analysis belongs in the Discussion. In particular, the use of “interestingly” in line 123 and noting sources of contamination in lines 134-136.
  • In regard to the figure:
    • Please make sure that your labels are correct.  They do not seem to be centered over the applicable bands.  Either that or please crop the gel to only include the labeled bands.
    • Please include the data from the protein identification.
    • Why were varying amounts of total protein used?  Why not standardize to the amount of negative control run?
    • Where do the tested allergens run?  In other words, why was a positive control not included?  Are antibodies available that are suitable to perform a Western blot for the allergens?
  • In regard to the table:
    • Please include additional footnotes to define the abbreviation ND and note which numbers were generated by an ELISA and which numbers were generated by the Lowry assay.
    • If your table breaks over multiple pages, repeating the column headings adds clarity for your readers.
    • Your results are summarized so clearly and concisely in the text, yet your table is rather unwieldy.  You may wish to make subdivisions in the table based on your groupings from the in-text descriptions.
  • In regard to the Discussion:
    • On line 155, the caveat is noted about statements made “in English”.  Out of curiosity, and given the availability of free translation tools online, could the remaining packaging be translated as well?
    • Line 177 finally describes why the 40C wash was performed.  This point is lost given its lack of introduction.  Further, is an 8-fold reduction in allergen amount sufficient to prevent detrimental health problems?  And where are the results of this assay shown?
    • Why are certain words underlined in lines 191-192?

Author Response

Reviewer 1 Reply.

Many thanks for your constructive and helpful comments on the draft paper.

  1. Purpose of the manuscript…… I have altered the introduction to make the purpose clear. The major purpose was to see if any NRL allergens were present in the material examined. But, given the expense of measuring NRL allergens, a secondary purpose was to see if a simple protein assay might be used to indicate the presence of NRL allergens. I have altered the discussion to make keep the appropriate balance of major and secondary aims of the work and added a short conclusion section
  2. Discussions of the protein measurement…… I have ensured that a common description of the protein method is used across the draft paper and indicated why we used this method
  3. Material and Methods section……..I have added appropriate information.
  4. Lines 64-77….. my apologies I have removed the publisher instructions.
  5. Materials….. have manufacturer information added now.
  6. Line 96. Centrifugation was at 8C. This has been added.
  7. The wash procedure has been moved to methods section.
  8. Relevance of detectable levels of allergens. I have added text in discussion.
  9. Moved any “discussion “ in the results section to the appropriate place (lines 123, 134-136.
  10. My apologies, I am having problems with the lane marking
  11. Additional information on the proteomic protein identification added.
  12. Varying absolute amounts of total protein were added to the gel as we have a dedicated fixed volume pippette for electrophoresis work and our operating procedure allow for addition of between 10-50 ug of total protein per lane……….. in hindsight it might have ben more sensible to add a constant amount of protein per lane.
  13. I have added the molecular weights of the NRL allergens to the text in the discussion.  Note that the purpose of the gel was not to see if NRL allergens were present, but to try and investigate  the high apparent protein levels in some extracts by electrophoresis, after excision of visible bands and proteomic identification. So we did not add a “positive control” of NRL allergens. Also  to add 14-40 microgrammes of the four NRL allergens would be very expensive.  Given that NRL allergens would be in the low nanogramme to picogramme amounts loaded on the gel we would not expect to have them visualised on an SDS gel stained with Coomassie blue stain.
  14. Additional footnotes have been added. I have split the original table in two tables reflecting the text descriptions.
  15. Statements in English. Doubtless some of the packaging could be translated into English. Howver as this refers to the N95/KN95 masks that were imported into the UK under government contract, any risks or hazard identification should be clearly identified on the packaging in English as a legal requirement.
  16. Wash experiment now appropriately mentioned in methods, results and discussion.
  17. Underlining removed

Reviewer 2 Report

Respected Authors,

The manuscript entitled "
Latex allergens in the ear straps of face coverings and masks in the UK: initial findings" covers a really important and practically relevant topic, which has a pontential impact beside the COVID-19 era. After a major revision, where some imprecise points are corrected, and the novelty can be highlighted, I recommend it for consideration for publication. 

In more details:

In the abstract, I recommend to modify the term "small study", simply to "study" 

Introduction: 

1.) The introduction is relatively short compared of the importance and significance of the topic. I would suggest to include the following aspects: 

Short, brief summery about the COVID-19 pandemic (definition of SARS-CoV2 and COVID-19, importance, impact), a more detailed introduction on medical supply shortage, higlighting how  innovative solutions helped the overcome on the urgent need of PPE-s, with a special focus what are the possible threats or disadvantages using these rapidly developed and produced equipment (including allergic reaction). I recommend to use the following sources: 

https://doi.org/10.1002/hpm.3009

https://doi.org/10.3390/ma13153363

https://doi.org/10.3390/polym12112703

https://doi.org/10.1177/0972063420935653

2.) The end of the introdcution should be re-written. I recommend to remove the part what initiated the study (the  report of two healthcare workers), and instead of that, it would be better to emphasize what is a goal and the novely of this study, compared to previous publications? 

Materials and Methods:

1.) The first paragraphs are from the Template (line 64-78). Remove it.
2.) Why the manufacturers or re-sellers are not indicated in case of the samples

The Results and clearly described. 

After the Discussion, I would recommend to include a Conclusion part, where the authors summarize their main findings and the most important take home messages and further plans.

Thank you 

Author Response

Reviewer 2 reply.

Thank you for your constructive comments on the draft paper.

Abstract

I have changed  “small study” to “study”.

Introduction

I have expanded  the introduction to highlight supply chain shortages, the spur to innovation etc.

I have retained sentence about the two health care workers as to me it is the reason why I as a part of an occupational health team became interested in the issue rather than our PPE experts or material scientists.  But I have clearly stated why the study was undertaken (its purpose), and the lack of any published data on latex in PPE during the pandemic.

Material and methods

Apologies for leaving part of the template in the draft. I have removed it.

The manufacturers/origins  are not indicated in the draft paper as there is a formal UK Parliamentary review of PPE provision during the pandemic.  As a government employee, I have to follow guidance concerning how far I can identify specific suppliers of PPE. However, I do not think this necessarily detracts from the draft paper.

I have added a short Conclusion at the end of the draft paper along the lines you suggest.

Round 2

Reviewer 2 Report

Thank you very much for the corrections.